# The Fuzzy Border between the Functional and Dysfunctional Effects of Beta-Amyloid: A Synaptocentric View of Neuron–Glia Entanglement

**DOI:** 10.3390/biomedicines11020484

**Published:** 2023-02-08

**Authors:** Francesca Fagiani, Tamas Fulop, Stefano Govoni, Cristina Lanni

**Affiliations:** 1Neuroimmunology Unit, Division of Neuroscience, Institute of Experimental Neurology (INSPE), IRCCS San Raffaele Hospital, 20132 Milano, Italy; 2Geriatric Division, Department of Medicine, Faculty of Medicine and Health Sciences, Research Center on Aging, Université de Sherbrooke, Sherbrooke, QC J1J 3H5, Canada; 3Department of Drug Sciences, Pharmacology Section, University of Pavia, 27100 Pavia, Italy

**Keywords:** beta-amyloid, synaptic activity, neurotransmitter release, microglia

## Abstract

Recent observations from clinical trials using monoclonal antibodies against Aβ seem to suggest that Aβ-targeting is modestly effective and not sufficiently based on an effective challenge of the role of Aβ from physiological to pathological. After an accelerated approval procedure for aducanumab, and more recently lecanemab, their efficacy and safety remain to be fully defined despite previous attempts with various monoclonal antibodies, and both academic institutions and pharmaceutical companies are actively searching for novel treatments. Aβ needs to be clarified further in a more complicated context, taking into account both its accumulation and its biological functions during the course of the disease. In this review, we discuss the border between activities affecting early, potentially reversible dysfunctions of the synapse and events trespassing the threshold of inflammatory, self-sustaining glial activation, leading to irreversible damage. We detail a clear understanding of the biological mechanisms underlying the derangement from function to dysfunction and the switch of the of Aβ role from physiological to pathological. A picture is emerging where the optimal therapeutic strategy against AD should involve a number of allied molecular processes, displaying efficacy not only in reducing the well-known AD pathogenesis players, such as Aβ or neuroinflammation, but also in preventing their adverse effects.

## 1. Introduction

The June 2021 announcement that the Alzheimer’s disease (AD) drug aducanumab, a monoclonal antibody directed against beta-amyloid (Aβ) fibrils and soluble oligomers, has been approved by the USA Food and Drug Administration to treat people with mild cognitive impairment (MCI) or mild-stage dementia, has provided a new impulse for the search of attainable targets in order to reverse, halt, or at least slow the progression of the disease. However, brain magnetic resonance imaging (MRI) abnormalities have been reported and related to the use of Aβ-targeting monoclonal antibodies, included aducanumab, in AD patients [1]. More recently, the multicenter, double-blind, phase 3 CLARITY AD trial reported that lecanemab, an antibody targeting Aβ soluble protofibrils, approved by the FDA through the accelerated procedure [2], reduced cognitive decline, as measured by the Clinical Dementia Rating-Sum of Boxes (CDR-SB), by 27% compared to placebo (an absolute difference of 0.45 points (change from baseline 1.21 for lecanemab vs. 1.66 with placebo, *p* < 0.001) in early AD patients (i.e., MCI patients and patients with mild dementia due to Alzheimer’s disease) [3]. However, a difference such as 0.45 on the 18-point CDR-SB scale has raised concerns about the clinical relevance of this minimal difference. Moreover, incidence of amyloid-related imaging abnormalities (ARIA) (i.e., adverse events associated with anti-Aβ antibodies), such as oedema (ARIA-E) or microhemorrhages (ARIA-H), has been observed in 21% of the lecanemab group [3]. Although the biological causes of the ARIA are still unknown, published hypotheses indicate that this phenomenon may be brought on by a combination of factors, including weakened vessel walls, increased cerebrovascular permeability due to Aβ clearance from neuritic plaques, and associated saturation of perivascular drainage [4]. Also, a role has been hypothesized for the inflammatory reactions associated with Aβ removal through monoclonal antibodies; indeed, alternative mechanisms not eliciting inflammatory reactions and not associated with microhemorrhages and angiopathy have been proposed [5]. The consequences of ARIA vary depending on the fact that monoclonal antibodies target different epitopes of Aβ or different species (monomers, oligomers, fibrils) or forms of the peptide (soluble or insoluble) [6]. Taking into account that, beyond its neurotoxic role, Aβ exerts key physiological functions, such as the neuromodulatory control of synaptic activity and neurotransmitter release from the presynaptic terminals, an antibody selectively binding and removing monomers may not be detrimental. Accordingly, by assessing the incidence of ARIA in clinical trials of anti-Aβ immunotherapy and comparing the incidence among different agents, a recent meta-analysis highlighted that cohorts treated with aducanumab displayed a significantly higher incidence of ARIA-effusion (E) and ARIA-hemorrhage (H) (30.7% and 30.0%) compared with other drugs [7].

Within this context, Aβ needs to be further settled in a more complex context, considering not only its accumulation, but also its biological effects within the different time frame of the disease course. It is important to mention that Aβ possesses several essential physiological functions, including (a) being part of the innate immune response as an antimicrobial peptide; (b) its protective role against brain injury; (c) potentially contributing to sealing the blood brain barrier during injury; (d) potentially being angiogenic depending on its form; and (e) potentially being antitumorigenic by promoting cell death [8,9].

In this review, we will provide a thorough understanding of the biological mechanisms underpinning the shift in the role of Aβ from physiological to pathological and the deviation from function to dysfunction, thereby facilitating a discussion of the border between activities impinging upon early, presumably reversible dysfunctions of the synapse and events trespassing the threshold of inflammatory self-sustaining glial activation leading to irreversible damage, mainly focusing on microglial function.

## 2. The Neuromodulatory Effect of Beta-Amyloid in Physiology

Clinical studies using animal models have widely demonstrated the importance of Aβ, a 4-kDa peptide derived from the sequential proteolytic cleavage of the amyloid precursor protein (APP) by β- and γ-secretase, in the progression of AD. In addition to its widely investigated role as one of the pathognomonic markers responsible for neurodegenerative processes, in the past fifteen years, significant advances regarding Aβ as an important synaptic regulator, inducing several functional and morphological synaptic changes and thus affecting age-related synaptic changes, have been made. These defects in synaptic activity are recognized as one of the earliest events in AD, preceding the deposition of Aβ plaques into the brain [10], and emphasize the role of Aβ in triggering earlier structural and functional perturbations of synaptic homeostasis in conditions not resulting in neurotoxicity [11]. These effects have been found to be differential due to the different variants, concentrations, and aggregation forms of Aβ peptides (i.e., monomers, oligomers, protofibrils, and fibrils) in the different experimental settings [12], as well as to the supplier-to-supplier and batch-to-batch variability of synthetic Aβ peptides [13]. Aβ has been demonstrated to act in a biphasic manner, exerting neuromodulatory/neuroprotective vs. neurotoxic effects, on the basis of its concentration and aggregation [14,15]. A functional interaction between Aβ and various neurotransmitter systems, including cholinergic, glutamatergic, GABAergic, catecholaminergic, and serotoninergic, has been discussed (on this topic, see [16]). When present at low concentrations (picomolar to low nanomolar), Aβ peptides positively affect neurotransmission and memory, whereas, in the high nanomolar–low micromolar range they exert a negative and neurotoxic effect on memory and synaptic plasticity. 

Of note, Aβ effects on neurotransmission may be responsible for early behavioral disturbances before the neurodegenerative phase. Indeed, the derangement of the neuromodulatory effect has been related to the pathological increase in Aβ levels and may trigger the perturbation of synaptic homeostasis and neurotransmission, thereby possibly contributing to the onset of “non-cognitive” symptoms of AD, usually referred as neuropsychiatric symptoms (NPS). They include pronounced symptoms, usually represented by apathy, agitation, phobias and anxiety, delusions, irritability, and sleep impairments. In AD, such behavioral signs have been indicated as contributors to a mild behavioral impairment construct [17], which includes sustained and impactful NPS preceding and predictive of incipient cognitive decline, and have been correlated to early synaptic dysfunction rather than to neurodegenerative processes. NPS, even if traditionally associated with frontotemporal dementia, have been suggested as predictive signs of incipient dementia, observable even before the onset of MCI. Within this context, tentative behavioral correlates of the Aβ-induced altered neurotransmission have been made. As an example, in murine models of Aβ amyloidosis, obtained by knocking-in a humanized Aβ sequence, before the onset of cognitive deficits, behavioral changes have been observed in association with non-cognitive, emotional domains [18]. Moreover, i.c.v. administration of the soluble Aβ 1–42 peptide in young adult male rats has been observed to induce motivational deficits, mainly depressive-like behavior (but not anxiogenic-like phenotype), decrease serotonin release in the cortex, and reduce the levels of neurotrophines, without affecting working memory [19,20].

According to Taragano et al. the mild behavioral impairment (MBI) syndrome identifies patients who are at risk of developing dementia whether or not they have cognitive symptoms, as well as includes a counterpart to MCI and a transitional condition between normal aging and dementia [21]. MBI score was found to be related to both global and striatal Aβ burden [22], and a very recent association has been observed between MBI and plasma Aβ42/Aβ40 in subjects with normal cognition or mild cognitive impairment [23]. In particular, in a small population of 86 cognitively intact elderly and 53 MCI subjects, lower plasma Aβ42/Aβ40 has been related to higher MBI score and affective dysregulation, but not with MBI areas associated with reduced drive/motivation or impulse dyscontrol [23]. Small sample sizes and the scarcity of independent studies continue to be a problem, nevertheless. Future research needs to investigate MBI associations with accumulating AD pathology [24], as well as to characterize which factors contribute to the progression from MBI to full-blown dementia. 

These speculative behavioral and clinical correlates could be helpful in identifying or recognizing changes in neurotransmitter activity as early prodromal signs of AD and as a potential target for pharmaceutical treatments, as suggested by David et al. [25]. Moreover, these observations may explain the limitations of current interventions, possibly enabling the preservation of Aβ physiological activity while counteracting its deposition.

Of note, this picture becomes even more articulated when taking into account that several other factors are critical in AD development and their prevention may therefore be a desirable strategy to slow the course and symptoms of AD. Among the variables not sufficiently explored and relevant, is gender. The incidence rates of AD are greater in women than in men, which is at least in part consistent with women’s survival to older ages. On the contrary, in their report on sex differences in AD, Guo et al. underscored the importance of sex-biased molecular pathways, including neuroinflammation and energy metabolism, and indicated the need for further attention and efforts to integrate multiomics data from different brain regions and cell types in examining the role of sex differences in AD [26]. In this context, data from the literature showed how, regarding synaptic plasticity in the hippocampus, neurons of females specifically respond to estradiol, whereas neurons of males are sensitive to androgens [27]. Evidence suggests that neuroactive steroids differentially control rodent learning and memory function. In the spontaneous alternation, passive avoidance, and Morris water maze tests, the memory deficits induced by acute Aβ 25–35 in young adult male mice were mitigated by the sulfated steroids pregnenolone sulfate and dehydroepiandrosterone sulfate, through the activation of nicotinic acetylcholine receptors sigma1 and α7 [28]. The neurosteroid modulation may improve memory by extending the lifespan of adult-born neurons, which may have positive effects on the progression of the disease [29]. Despite data on the effects of sex steroids on rodent and human memory, however, a wide range of outcomes has been reported in terms of improvement, reduction, or no effect depending on the disease stage, subject gender, study design, mechanism of delivery, type of memory assessed, and steroid dosage [30]. Hence, studies need to be improved in order to provide answers on a model fully recapitulating AD and the therapeutic time frame of intervention, ameliorate molecule efficacy and safety, and standardize validated methodologies.

## 3. Beta-Amyloid as Endogenous Regulator on Synaptic Activity and Neurotransmitter Release

An increasing amount of literature supports the concept that soluble Aβ regulates important physiological processes, such as synaptic plasticity and memory, to serve as an essential synaptic regulator. Puzzo et al. demonstrated that synaptic plasticity and memory were positively altered when hippocampal neurons were exposed to low concentrations (i.e., picomolar–low nanomolar) of Aβ 1–42 [15]. On the other hand, exposure to higher concentrations (high nanomolar–low micromolar) resulted in a neurotoxic effect. In particular, Aβ 1–42 controlled long-term potentiation (LTP), the electrophysiological correlate of learning and memory, in a biphasic or hormetic manner [15]. As a result, picomolar concentrations of Aβ 1–42 enhanced LTP enhancement at the synapses between Schaffer collateral fibers and CA1 neurons, with a maximum effect occurring at 200 pM, whereas nanomolar concentrations of Aβ 1–42 caused an impairment of LTP. In addition, Gulisano et al. found that mouse CA1 pyramidal neurons exposed to 200 pM low-molecular-weight oligomeric Aβ 1–42 showed an increase in the frequency of small excitatory postsynaptic currents and a decrease in paired-pulse facilitation [31]. Further evidence that low concentrations of oligomeric Aβ 1–42 stimulate neurotransmitter release from the presynaptic terminals comes from the observation of an increased number of docked vesicles at presynaptic terminals. It is noteworthy that these effects were not seen when pyramidal neurons were exposed to 200 pM Aβ 1–40 oligomers. These data highlight how the effects of soluble Aβ on synaptic plasticity and memory depend not only on the concentration of the peptide but also on the various isoforms and the aggregation status of Aβ. Another important factor to take into account is the duration of the exposure to the peptide. In this regard, Koppensteiner et al. demonstrated that synaptic plasticity in mouse hippocampal neurons and contextual memory in mice were stimulated by short-term exposure (minutes) to a picomolar concentration (200 pM) of oligomeric Aβ 1–42, whereas prolonged (hour-long) exposures to 200 pM Aβ 1–42 led to a decrease in such parameters [32]. Given that Aβ levels fluctuate throughout the day, it is crucial to keep in mind that dynamic Aβ alterations physiologically occur in the brain. Accordingly, soluble Aβ levels have been found to exhibit strong daily oscillations with a distinct 24-h period in both mouse hippocampal interstitial fluid and human CSF [33,34], indicating the existence of physiological circadian patterns regulating fluctuations of CSF Aβ levels. Notably, Huang et al. showed that the typical CSF Aβ dynamics are diminished to a flat line with aging and Aβ accumulation, likely contributing to AD [34].

Furthermore, data from the literature indicate that Aβ controls neurotransmitter release from presynaptic terminals without inducing neurotoxic effects. In this regard, endogenous Aβ has been shown to play a crucial role in controlling synaptic vesicle release, without affecting postsynaptic function. In particular, in in vitro mouse hippocampal cells, an increase in the synaptic vesicle release, as well as in neuronal activity, has been observed when endogenous Aβ levels were elevated by the inhibition of its extracellular breakdown [35]. Aβ may directly interact with presynaptic proteins that are crucial in coordinating the neurotransmitter release machinery, thereby altering numerous synaptic vesicle cycle events, including vesicle docking and fusion, which are essential for synaptic vesicle exocytosis as well as vesicle recycling and recovery in neurons (for a comprehensive review on the topic, see [36]). Beginning with low levels of Aβ monomers, fusion stimulation and endocytosis inhibition may promote synaptic reinforcement. The exocytosis inhibition may prevail with further increases of Aβ and the onset of aggregation events, thus impairing nerve terminals, particularly those firing frequently, and accompanied by an inhibition of release that results in a more widespread synaptic failure. Indeed, the neuromodulatory effect of Aβ has been found to be crucial in maintaining the right balance between the various neurotransmitter systems, including cholinergic, glutama-tergic, GABAergic, catecholaminergic, and serotoninergic [16]. Several in vitro and in vivo models revealed that Aβ regulates, in different brain regions, the cholinergic control of neurotransmitter release, depending on its concentration and aggregation status (for a comprehensive review on the topic, see [36]). In particular, pM-nM concentrations of Aβ 1–40 stimulated both excitatory (aspartate and glutamate) and inhibitory amino acid (GABA and glycine) release in response to nicotine, while, with higher concentrations of the peptide, a reduction in the nicotine-evoked release of glutamate and aspartate occurred [37,38]. The need for maintaining healthy Aβ concentrations is highlighted by this dual effect of Aβ pathology, indicating that Aβ accumulation causes a shift in which the neural network changes from being primarily excitatory to becoming increasingly inhibitory as the pathology progresses. 

Further research on the interaction between Aβ and the presynaptic release mechanism may yield pertinent information given the important role that Aβ plays at presynaptic terminals as well as its effects on neurotransmitter release. Additional complexity derives from the observations that Aβ affects intracellular kinase signaling (e.g., calpain-cyclin-dependent kinase 5 and Ca^2+^/calmodulin-dependent protein kinase IV) [39,40], consequently affecting the presynaptic terminal’s fine-tuning of synaptic vesicle dynamics. Which downstream pathways are involved, how are they temporally controlled, and how endogenously produced Aβ (comprising multiple isoforms and molecular conformations) influences synaptic activity in normal and non-transgenic brain circuits need to be better understood. A detailed understanding of the molecular mechanisms driving the switch from function to dysfunction and the change of the Aβ role from physiological to pathological is hampered by such a constraint. 

## 4. Neuron–Glia Interaction: From Synaptic Regulation to Dysregulation

Evidence from the literature refers to AD as a synapse disease in which astrocyte and microglia activities, as well as pre- and postsynaptic processes, gradually decline or change. Synaptic integration between neurons, astrocytes, and microglia ensures optimal brain function and cognitive performance. When Aβ accumulates, the synapse is pushed from its healthy equilibrium toward a pathological state, and similar alterations in astrocyte and microglia function also participate in this transition [41]. In the early stages of AD, astrocytes and microglia act protectively and attempt to correct aberrant synaptic transmission by taking part in the Aβ clearance and the compensatory production of functional proteins. 

Accordingly, astrocytes use perisynaptic processes to maintain strong connections with neural synapses, and are crucial for neurophysiological signaling, recycling neurotransmitters, preserving tissue ion homeostasis, and controlling synaptic transmission by gliotransmitter release [42]. Microglia, by secreting cytokines and expressing enzymes, serve as the main regulators of neural plasticity [43] throughout development and adulthood, thus controlling the process known as “synaptic pruning” to remove inactive synaptic connections and maintain functional synapses [44]. Astrocytes drive microglia to synapses that have undergone complement pathway pruning [45], thereby affecting microglia–neuron interactions and the delivery of microglia-induced release of neurotoxic and neurotrophic substances to neurons. In particular, early astrocyte reactivity has been proposed to be protective against AD pathogenesis, based on data related to an increase in Aβ clearance [46], or the upregulation of proteins important for neurophysiology. As an example, astrocytes are mostly protective during the early stages of disease progression because they attempt to correct abnormalities in K^+^ homeostasis by upregulating Kir4.1 expression close to areas with severe Aβ pathology [47]. In more advanced stages of AD, astrocytes stop serving a protective role and shift toward a pro-inflammatory profile. Reactive astrocytes are intimately related to Aβ plaques and neurofibrillary tangles, and their specific role may change over the course of the disease, showing modifications in the transcriptional signature and changes of their signaling, interactions with pathogenic protein aggregates, and impairments of metabolism and synaptic function, with altered release of gliotransmitters [48]. As an example, GABA content was observed to be highly released by astrocytes in the hippocampus, primarily at later stages of AD progression near Aβ plaques [49]. Furthermore, astrocytic metabolism has been shown to adapt to amyloid plaques in vitro, with changes to glycolysis and mitochondrial activity as well as the activation of several intracellular pathways that result in inflammation, oxidative stress, and calcium dysregulation [50].

## 5. Microglia

While mounting evidence indicates that neuroinflammation plays a crucial role in AD, the impact of the microglia activation in AD onset and progression remains a matter of debate. It has been speculated that, while, in the early stage of the disease, microglia exert a neuroprotective effect by phagocytosing Aβ [51], in late-stage AD, chronic stimulation of microglia due to the AD environment pushes the microglial transition from a homeostatic to an increasingly pro-inflammatory state, with a detrimental impact on tissue homeostasis and, in particular, on synaptic function (Figure 1). As a consequence, abnormal microglial activation or dysfunction may have a significant impact on a number of signaling pathways, molecular functions, and interactions with neurons and astrocytes [52]. Notably, pre-plaque microglial activation has been seen in AD animal models [53], as well as in the prodromal stage of AD, as assessed by using 18F-DPA-714 together with amyloid imaging (PiB-PET) in a cohort of patients with AD at both prodromal and dementia stages [51], thus suggesting microglia-driven neuroinflammation as an early event in AD. Therefore, the identification of relevant changes in the microglial profile throughout the course of AD is emerging as a crucial aspect for shedding light on the molecular mechanisms of microglia regulation and for characterizing the activation phenotype of plaque-associated microglial cells and their differences from microglia distant from Aβ plaques. In this regard, major advances in understanding microglia activity in AD come from studies profiling the microglia transcriptome in AD preclinical models and *post-mortem* human brains, showing a wide spectrum of distinct microglial activation states [54,55]. In this regard, a unique microglial signature exclusively present in AD, i.e., the “disease-associated microglia” (DAM), was first described in proximity to Aβ plaques in 6 month-old AD mice [56], and then confirmed in models of tau pathology and *post-mortem* human tissues from AD patients [57]. In particular, Karen–Shaul et al., identified two clusters of DAM, clusters II (4.2%) and III (2.8%), whose transcriptional profile was characterized by the downregulation of microglial homeostatic genes, such as the purinergic receptors P2ry12/P2ry13, Cx3cr1, and Tmem119, and upregulation of key AD risk factor genes, including ApoE, Ctsd, Lpl, Tyrobp, and Trem2 [56]. Moreover, gene set enrichment analysis of DAM-specific genes showed marked activation of lysosomal/phagocytic pathways, endocytosis, and regulation of the immune response [56]. Notably, such DAM-phagocytic cells have been reported to express high levels of phagocytic and lipid metabolism pathways and to be spatially located in close proximity to Aβ, suggesting their involvement in Aβ phagocytosis. A strong overlap between Lpl-positive microglia, predominantly around Aβ plaques, has been also described in AD *post-mortem* brain samples [56]. The growing body of evidence based on the novel research tools and experimental approaches highlights the heterogeneity of the microglial population, as well as of their function [58,59] and the complex relationship of these microglia, metabolism, and Aβ in the synaptic dysfunction [60]. 

Among the evidence indicating microglial heterogeneity, Plesher et al. demonstrated relevant morphological and electrophysiological changes exclusively in Aβ plaque-associated microglia, compared to microglia distant from Aβ plaques, in the TgCRND8 mouse model of AD, implicating that the plaque microenvironment differentially affects microglial ion channel expression [61].

Recent studies also revealed new roles for microglia in modulating synaptic communication. In particular, Gabrielli et al. investigated the role of Aβ contained in vesicles released by microglia and the consequent implications of vesicular motion on alteration of synaptic plasticity and synapses decrease in the entorhinal–hippocampal circuitry, an important, vulnerable, and severely impaired area in AD patients [62]. These results are intriguing taking into account that in the first stages of AD an unexpected pathological role for microglia has been involved [63]. Extracellular vesicles are membrane vesicles represented by ectosomes/microvesicles (formed at the plasma membrane) or exosomes (generated in the endocytic compartment), divided into small (≤100–200 nm-diameter) and large (>200 nm-diameter) vesicles based on size, density, and biochemical content [64] and able to hold and transport cellular components and pathogenic proteins, such as Aβ [65,66]. In this study, Gabrielli et al. investigated the capability of large EVs (ectosomes) carrying Aβ 1–42 (Aβ-EVs) to reduce synaptic plasticity and spread synaptic dysfunction by moving near the axon surface, thus contributing to initial synaptic dysfunction. In particular, a picture for the early stages of AD, where Aβ accumulates in specific brain areas, is first absorbed by microglia, and is subsequently re-secreted in toxic form together with EVs, has been proposed. In particular, the authors found that large microglial Aβ-EVs affected synaptic plasticity and, once injected into the mouse brain, spread LTP impairment along the entorhinal–hippocampal circuitry. Of note, differently formed free oligomeric Aβ42 packaging into EVs makes Aβ capable of propagating synaptic dysfunction and able to be effective at a lower concentration compared to free soluble oligomeric Aβ (9 nM active concentration of EV-associated Aβ 42 versus 200 nM of free Aβ 1–42). Moreover, large EVs containing Aβ have been detected in the CSF of AD patients [67] and their synthesis by microglia is related to early brain damage in prodromal AD, indicating that endogenously produced large microglial EVs may be involved in the development of AD. Additionally, elevated levels of Aβ 1–42 in neuronal EVs have been detected in the blood of prodromal AD patients [68]. Proteins, lipids, and nucleic acids found in EVs could therefore serve as cutting-edge prospective biomarkers to track the progression of AD as well as possible targets for novel therapeutic approaches. Additionally, non-coding RNAs may contribute to the derangement of microglia and astrocytes in a variety of neurodegenerative illnesses [69] by blocking miRNA repression on target proteins. This rationale could lead to the development of new prospective therapeutic options based on the non-coding RNA–glial cell axis.

## 6. Concluding Remarks

According to the experience of clinical trials with a variety of prospective drugs, the results of recent outcomes using monoclonal antibodies against Aβ seem to suggest that Aβ-targeting is ineffective if it is not coupled with an effective challenge to the conversion of the Aβ role from physiological to pathological. One anti-amyloid antibody (aducanumab) was recently approved by the FDA as a novel AD disease-modifying medication through the accelerated approval procedure. Even taking into account the prior attempts with different monoclonal antibodies, there is still ambiguity regarding the actual efficacy and safety of aducanumab, and both academic institutions and pharmaceutical businesses are actively looking for novel treatments [70,71]. There may be several reasons related to the failure of the different therapeutic approaches to counteract AD. The rationale behind the therapeutic approaches, which in some cases is based on theoretical assumptions and data from animal models [72], the complexity of disease/syndrome pathogenesis, the timing of therapeutic intervention (which should anticipate the natural history of the disease), and the phenotypic and molecular variability of AD [73], which impact and diversify the responsiveness to treatments, are just a few of the possible causes of the failure of AD therapeutic approaches. 

Overall, these findings imply that multiple allied molecular mechanisms should be involved in the ideal therapeutic approach against AD, demonstrating efficacy not only in inhibiting the well-known players in AD pathogenesis, such as Aβ, neuroinflammation, and others, but also in preventing their pathological effects. In the research for novel investigational drugs to treat the illness, rare mutations with potential protective effects against AD, recently found through genetic studies, have inspired this field [74]. In this regard, Catania et al. developed an innovative strategy for AD by intranasally delivering an anti-amyloidogenic six-mer peptide (Aβ 1–6A2V), in turn derived from the natural genetic variant of Aβ (AβA2V) characterized to have anti-amyloidogenic properties. In a mouse model of AD, this strategy was successful in preventing the accumulation of wild-type Aβ and avoiding the synaptic damage brought on by amyloidogenesis, and actually may be included in the class of “amyloid β-targeted peptide inhibitors” [75]. 

It is also worth remembering that the emerging complexity of the picture underscores the importance of better defining the hierarchy and time-dependent involvement of the various pathways as they interact with the aging process. Such a process, as well as the full dialogue between peripheral structures and the brain, have not yet been studied systematically due to their complexity. Furthermore, the number of papers supporting a lifestyle correlation that may prevent cognitive decline is steadily increasing, implying that in taking advantage of the new research tools, we should directly face the complexity of pathological cognitive decline and that a personalized multimodal intervention should be the foundation of all novel therapeutic approaches.

## Figures and Tables

**Figure 1 biomedicines-11-00484-f001:**
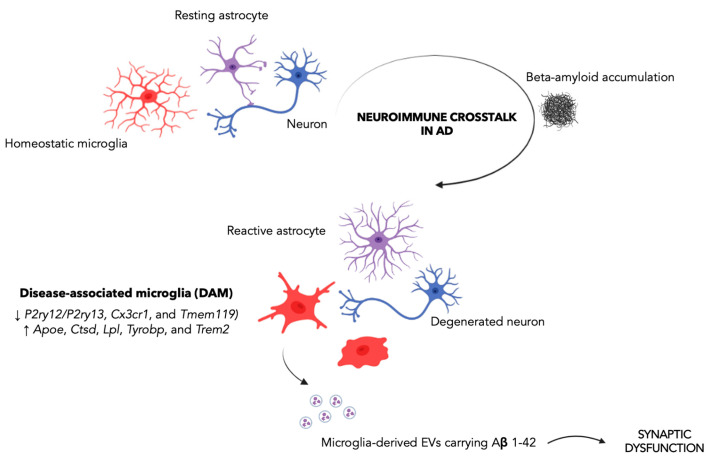
The disease-associated microglia as a driver of synaptic dysfunction in Alzheimer’s disease. The neuroimmune crosstalk between astrocytes, microglia, and neurons plays a crucial role in AD. In particular, it has been proposed that, in the early stage of the disease, microglia exert a neuroprotective effect by phagocytosing Aβ. Instead, in late-stage AD, chronic stimulation of microglia promotes microglia transition from a homeostatic to an increasingly pro-inflammatory state, with a negative impact on synaptic function. Specifically, large EVs (ectosomes) containing Aβ 1–42 have been reported to impair synaptic plasticity, thereby contributing to initial synaptic dysfunction. The Figure was created with BioRender.com.

## Data Availability

Not applicable.

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
