# Peer review of "The Fuzzy Border between the Functional and Dysfunctional Effects of Beta-Amyloid: A Synaptocentric View of Neuron–Glia Entanglement"

_biomedicines, 2023, doi:10.3390/biomedicines11020484_

Round 1

Reviewer 1 Report

The authors present an interesting review of a very important problem: Not only pathological, but also physiological role of amyloid-b in a brain tissue. However, in the current form the title of the manuscript does not match the content. In the title of the manuscript the authors proposed is emphasized on synaptocentric view of neuron-glia entanglement; Hovewer, in the manuscript the main discussion is in context of microglia. Thus, I hightly recommend the authors to discuss more about the neuron-glia interactions and synapses in the text of the manuscript, or at least to change the title of the manuscript.

Besides, there are the minor comments:

Line 23: the excessive 'of'

Line 174: not 'play a part', it's better to say 'play a role'

Line 306: If the Figure 1 was made using some on-line resources, such as Biorender.com for example, it should be pointed in the figure legend.

Author Response

We thank the Reviewer for the comments and suggestions, that allowed us to improve the paper.

Regarding the issues, we have modified and/or integrated the text (as highlighted in yellow) as follows (please see the attachment):

  • The authors present an interesting review of a very important problem: Not only pathological, but also physiological role of amyloid-b in a brain tissue. However, in the current form the title of the manuscript does not match the content. In the title of the manuscript the authors proposed is emphasized on synaptocentric view of neuron-glia entanglement; Hovewer, in the manuscript the main discussion is in context of microglia. Thus, I hightly recommend the authors to discuss more about the neuron-glia interactions and synapses in the text of the manuscript, or at least to change the title of the manuscript.

ANSWER: According the Reviewer observation, we modified the text in the section “Neuron-glia interaction: from synaptic regulation to dysregulation” and integrated the references as follows: “Evidence from literature refers to AD as a disease of the synapse, where pre- and postsynaptic processes, as well as astrocyte and microglia functions, progressively deteriorate/change. Neurons, astrocytes, and microglia integrate at the synapse to guarantee healthy brain functioning and cognitive performance. When Aβ accumulates, the syn-apse is pushed from its healthy equilibrium toward a pathological state and similar alterations in astrocyte and microglia function also participate to this transition [36]. In the early stages of AD, astrocytes and microglia act protectively and attempt to correct ab-errant synaptic transmission by taking part in the Aβ clearance and the compensatory production of functional proteins.

Accordingly, astrocytes use perisynaptic processes to maintain strong connections with neural synapses, and are crucial for neurophysiological signaling, recycling neurotransmitters, preserving tissue ion homeostasis, and controlling synaptic transmission by gliotransmitter release [37]. Microglia, by secreting cytokines and expressing enzymes, serve as the main regulators of neural plasticity [38] throughout development and adulthood, thus controlling the process known as "synaptic pruning" to remove inactive synaptic connections and maintain functional synapses [39]. Astrocytes guide microglia to synapses that have been pruned via the complement pathway [40], thus influencing the interaction of microglia with neurons, and thereby indirectly affecting the delivery of neurotoxic and neurotrophic factors to neurons released by microglia. In particular, early astrocyte reactivity has been proposed to be protective against AD pathogenesis, based on data related to an increase in Aβ clearance [41], or the upregulation of proteins important for neurophysiology. As an example, astrocytes are mostly protective during the early stages of disease progression because they attempt to correct abnormalities in K+ homeostasis by upregulating Kir4.1 expression close to areas with severe Aβ pathology [42]. In more advanced stages of AD, astrocytes stop serving a protective role and shift toward a pro-inflammatory profile. Reactive astrocytes are intimately related to Aβ plaques and neurofibrillary tangles, and their specific role may change over the course of the disease showing modifications in the transcriptional signature and changes of their signaling, interactions with pathogenic protein aggregates, and impairments of metabolism and synaptic function, with altered release in gliotransmitters [43]. As an example, GABA content was observed to be highly released in hippocampus by astrocytes primarily at later stages of AD progression near Aβ plaques [44]. Furthermore, astrocytic metabolism has been shown to adapt to amyloid plaques in vitro, with changes to glycolysis and mitochondrial activity as well as the activation of several intracellular pathways that result in inflammation, oxidative stress, and calcium dysregulation [45].

  • Besides, there are the minor comments:
    1. Line 23: the excessive 'of'
    2. Line 174: not 'play a part', it's better to say 'play a role'
    3. Line 306: If the Figure 1 was made using some on-line resources, such as Biorender.com for example, it should be pointed in the figure legend.

ANSWER: We modified the text as suggested by the Reviewer

Reviewer 2 Report

In the current review the authors with regard to the efficacy and side effects of the recently approved monoclonal antibody aducanumab, for Alzheimer’s disease (AD), review the literature focusing on neuroimmune crosstalk between astrocytes, microglia and neurons but make a clever approach. The authors highlight the complexity of AD and the need for further studies; they also analyze the reasons leading to the failure of therapeutic approaches and propose a personalized multimodal intervention giving emphasis on the timing of the intervention. The manuscript is interesting, very well organized, well written in a simplified and comprehensible manner, and fits in the scope of the journal. The scheme is simple and helpful.   English is fine, but very few mistakes to correct.

Author Response

In the current review the authors with regard to the efficacy and side effects of the recently approved monoclonal antibody aducanumab, for Alzheimer’s disease (AD), review the literature focusing on neuroimmune crosstalk between astrocytes, microglia and neurons but make a clever approach. The authors highlight the complexity of AD and the need for further studies; they also analyze the reasons leading to the failure of therapeutic approaches and propose a personalized multimodal intervention giving emphasis on the timing of the intervention. The manuscript is interesting, very well organized, well written in a simplified and comprehensible manner, and fits in the scope of the journal. The scheme is simple and helpful. English is fine, but very few mistakes to correct.

ANSWER: We thank the Reviewer for the positive comments. As suggested, our paper has been revised by an English mother tongue in order to make the text more fluent.

Reviewer 3 Report

The manuscript examines the function of beta amyloid protein under physiological and pathological conditions. In particular, neuromodulatory/neuroprotective effects, and possible neurotoxic effects, are considered. The topic is complex and very broad, but the authors have given a precise slant to the work making the manuscript interesting and quite original. The discussion, also based on animal models of amyloidosis and the role of A beta proteins in relation to neuronal and microglial regulatory functions, appears well argued and thorough.

Considering the possible gender differences of the influence what gender-specific effects can be highlighted? What is the synaptic modulatory role of estrogenic and androgenic sex neurosteroids, such as estradiol and testosterone, in the ability of A beta to influence neuronal memory and possibly cognition?

Could therapeutic approaches based on neurosteroid modulation in conditions such as MCI be hypothesized to address the synaptic effects of amyloidosis?

Author Response

We thank the Reviewer for the comments and suggestions, that allowed us to improve the paper.

Regarding the issues, we have modified and/or integrated the text (as highlighted in blue) as follows (see the attachment):

  • The manuscript examines the function of beta amyloid protein under physiological and pathological conditions. In particular, neuromodulatory/neuroprotective effects, and possible neurotoxic effects, are considered. The topic is complex and very broad, but the authors have given a precise slant to the work making the manuscript interesting and quite original. The discussion, also based on animal models of amyloidosis and the role of A beta proteins in relation to neuronal and microglial regulatory functions, appears well argued and thorough. Considering the possible gender differences of the influence what gender-specific effects can be highlighted? What is the synaptic modulatory role of estrogenic and androgenic sex neurosteroids, such as estradiol and testosterone, in the ability of A beta to influence neuronal memory and possibly cognition?Could therapeutic approaches based on neurosteroid modulation in conditions such as MCI be hypothesized to address the synaptic effects of amyloidosis?

ANSWER: According the Reviewer observation, we modified the text in the section “The neuromodulatory effect of beta-amyloid in physiology” and integrated the references as follows: “Noteworthy, this picture becomes even more articulated when taking into account that several other factors are critical in AD development and their prevention may therefore be a desirable strategy to slow the course and symptoms of AD. Among the variables not sufficiently explored and relevant, one is gender. The incidence rates of AD are greater in women than in men, which is at least in part consistent with women survival to older ages. On the other hand, in their report on sex differences in AD, Guo et al.  underscored the importance of sex-biased molecular pathways including neuroinflammation and energy metabolism and indicated the need of further attention and efforts to integrate multiomics data from different brain regions and cell type in examining the role of sex differences in AD [26]. In this context, data from literature showed how, regarding synaptic plasticity in the hippocampus, female neurons specifically respond to estradiol, whereas male neurons are sensitive to androgens [27]. Evidence suggests that neuroactive steroids differentially control rodent learning and memory function. In the spontaneous alternation, passive avoidance, and Morris water maze tests, the memory deficits induced by acute Aβ 25-35 in young adult male mice were mitigated by the sulfated steroids pregnenolone sulfate and dehydroepiandrosterone sulfate, through the activation of nicotinic acetylcholine receptors sigma1 and a7 [28]. The neurosteroid modulation may improve memory by extending the lifespan of adult-born neurons, which may have positive effects on the progression of the disease [29]. Despite data on the effects of sex steroids on rodent and human memory, however, a wide range of outcomes in terms of improvement, reduction, or no effect has been reported depending on the disease stage, subject gender, study design, mechanism of delivery, type of memory assessed, and steroid dosage [30]. Hence, studies need to be improved to answer on a model fully recapitulating AD, the therapeutic time frame of intervention, and to ameliorate molecule efficacy and safety, as well as to standardize validated methodologies”.

Round 2

Reviewer 1 Report

The authors added the significant discussion considering the role of different glial cells in the synaptic homeostasis. In my opinion, the manuscript should be accepted.